

# Behavioral correlates of semi-zygodactyly in Ospreys (*Pandion haliaetus*) based on analysis of internet images

Diego Sustaita[1], Yuri Gloumakov[2], Leah R. Tsang[3,4] and
Aaron M. Dollar[2]

[1] Department of Biological Sciences, California State University, San Marcos,
San Marcos, CA, USA
[2] Department of Mechanical Engineering and Materials Science, Yale University,
New Haven, CT, USA
[3] Department of Zoology, Environmental and Rural Sciences, University of New England,
Armidale, NSW, Australia
[4] Ornithology Collection, Australian Museum Research Institute, Sydney, NSW, Australia

## ABSTRACT

Ospreys are renowned for their fishing abilities, which have largely been attributed to their specialized talon morphology and semi-zygodactyly—the ability to rotate the fourth toe to accompany the first toe in opposition of toes II and III. Anecdotal observations indicate that zygodactyly in Ospreys is associated with prey capture, although to our knowledge this has not been rigorously tested. As a first pass toward understanding the functional significance of semi-zygodactyly in Ospreys, we scoured the internet for images of Osprey feet in a variety of circumstances. From these we cross-tabulated the number of times each of three toe configurations (anisodactylous, zygodactylous, and an intermediate condition between these) was associated with different grasping scenarios (e.g., grasping prey or perched), contact conditions (e.g., fish, other objects, or substrate), object sizes (relative to foot size), and grasping behaviors (e.g., using one or both feet). Our analysis confirms an association between zygodactyly and grasping behavior; the odds that an osprey exhibited zygodactyly while grasping objects in flight were 5.7 times greater than whilst perched. Furthermore, the odds of zygodactyly during single-foot grasps were 4.1 times greater when pictured grasping fish compared to other objects. These results suggest a functional association between predatory behavior and zygodactyly and has implications for the selective role of predatory performance in the evolution of zygodactyly more generally.

# INTRODUCTION

Ospreys (Accipitriformes: Pandionidae: *Pandion haliaetus*) feed virtually exclusively on fish (accounting for ∼99% of their diet) that they take from the water (*Bierregaard et al., 2016*). They are able to achieve substantial prey-capture success rates for a predator (up to 82%; *Bierregaard et al., 2016*), despite the difficulties inherent when plunge-diving feet-first to capture fish. This ability is afforded by their unique pedal anatomy, compared

Corresponding author
Diego Sustaita, dsustaita@csusm.edu

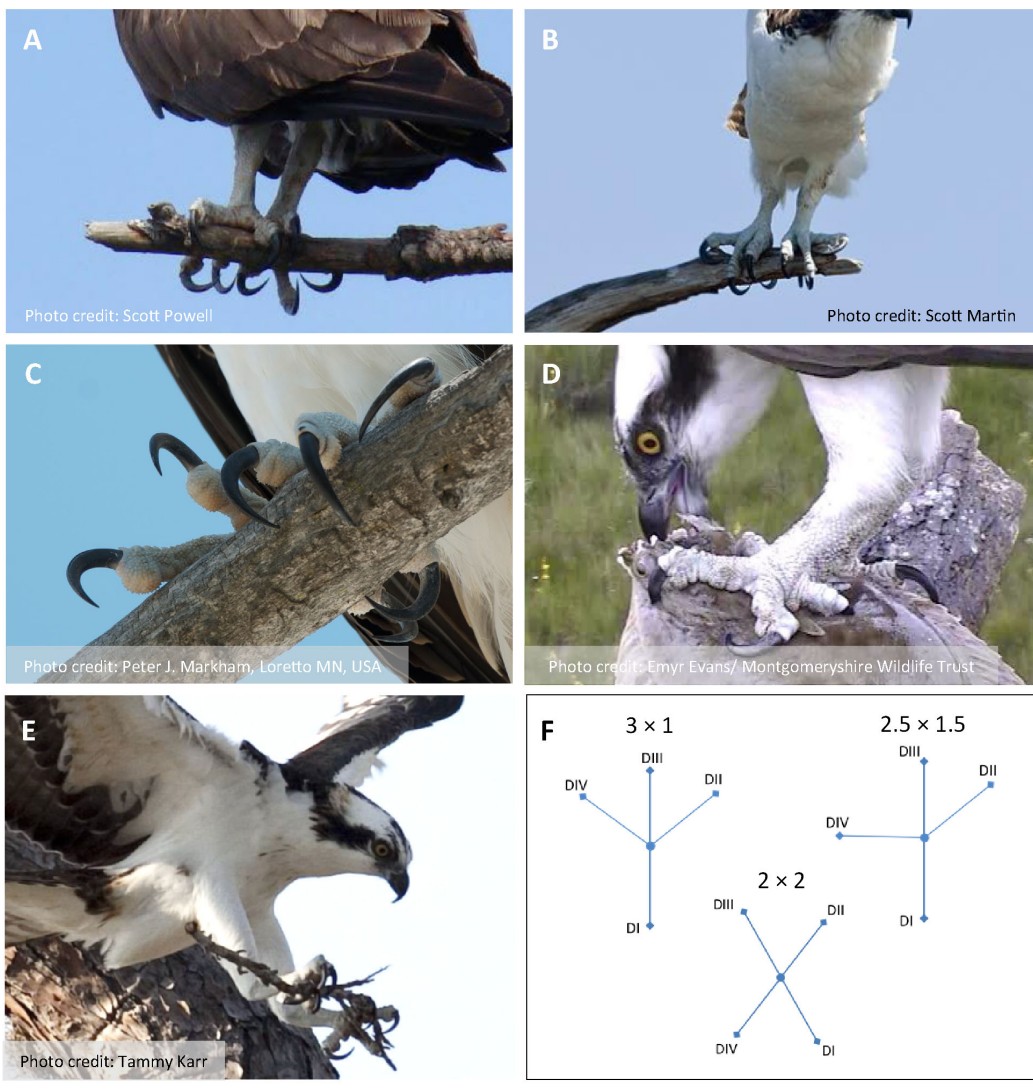

**Figure 1 Photos of Ospreys showing grasping scenarios and representative object types and sizes.**
(A) Perched, grasping a tree branch (small) with a 2 × 2 configuration in the left foot and a 3 × 1 configuration in the right foot (photo by Scott Powell). (B) Perched, grasping a tree branch (small) with a 2.5 × 1.5 configuration in the left and right foot (photo by Scott Martin). (C) Perched, grasping a tree branch (medium) with a 3 × 1 configuration in the left foot and a 2.5 × 1.5 configuration in the right foot (photo by Peter J. Markham). (D) Perched, grasping (single-footed) a fish (large), with a 2 × 2 configuration in the left foot (photo by Emyr Evans/Montgomeryshire Wildlife Trust). (E) Flying, grasping (dual-footed) a twig (small) using a 2 × 2 configuration in the left and right foot (photo by Tammy Karr). (F) Schematic diagrams of a left foot showing foot types scored in A–E.

to other birds of prey. Among these attributes is the ability to rotate the fourth toe (digit IV) antero-posteriorly, and toggle between anisodactyl (digits II–IV face anteriorly; digit I posteriorly) and zygodactyl (digits II and III face anteriorly; digits I and IV face posteriorly) toe arrangements (*Shufeldt, 1909*; *Jollie, 1976*, *1977*; *Raikow, 1985*; *Polson, 1993*; *Ramos & Walker, 1998*; *Tsang & McDonald, 2018*) (Fig. 1F). The ability to facultatively shift from anisodactyly to zygodactyly (i.e., semi-zygodactyly; *Raikow, 1985*) is thought to enhance their extreme grasping capabilities. For instance, previous

researchers have proposed that the facultative zygodactyl arrangement in predatory birds, such as owls (Strigiformes) and Black-shouldered Kites (Accipitriformes: Accipitridae: *Elanus axillaris*) (*Tsang, 2012*), provides advantages for distributing the toes (and prey-contact surface area) more symmetrically (*Payne, 1962*; *Goslow, 1972*), as well as for generating greater grip strength (*Ward, Weigl & Conroy, 2002*; *Einoder & Richardson, 2007*). Both of these advantages ostensibly pertain to the Osprey, which grasps evasive, slippery fish from above by plunge-diving to capture prey about half a meter below the surface of the water (*Polson, 1993*).

That Ospreys can reverse their fourth toe to assume a zygodactyl foot configuration is fairly well known (*Johnsgard, 1990*; *Olsen, 1995*; *Ferguson-Lees & Christie, 2001*; *Bierregaard et al., 2016*). However, it is not abundantly clear specifically when and how they employ one toe configuration over the other. Casual observations of ospreys captured in photographs reveal that the zygodactyl configuration is often assumed during perching as well as when clutching fish. Thus, the extent to which Ospreys preferentially use zygodactyly for grasping prey, although perfectly reasonable, is not explicitly clear. Furthermore, it is unclear specifically how the change in toe configuration is controlled. Ospreys possess gross anatomical peculiarities that are presumed to be associated with semi-zygodactyly. These include a relatively long digit IV that is semi-reversible, the absence of a membrane between digits III and IV (*Tsang, 2012*), and claws of near equal length across all toes (*Hudson, 1948*; *Jollie, 1976*, *1977*). In regards to the underlying bones, the inner trochlea of the distal tarsometatarsus is comparatively more developed than in other accipitriforms, which might afford the second digit a relatively greater range of motion, and the shape of the outer trochlea seems to reflect the "wide lateral movement" exhibited by digit IV (*Jollie, 1976*, *1977*). Finally, their comparatively well-developed M. lumbricales and M. abductor digiti IV muscles (*Hudson, 1948*) reflect their keen abilities to abduct and reverse digit IV, unlike other raptors. However, the extent to which Ospreys are able to reposition digit IV voluntarily, or if such repositioning is mechanistically coupled with other hindlimb or digital movements, is unclear.

As part of a larger project aimed at understanding the anatomy, control, and functional significance of semi-zygodactyly in Ospreys, we first set out to examine the behavioral correlates of semi-zygodactyly. We approached this by quantifying foot use behaviors captured in digital images and videos publicly available on the internet, using a methodology similar to *Allen et al.'s (2018)* image-based study of foot lateralization in Ospreys, although derived independently. We used data gleaned from these images specifically to test for associations among toe configurations, grasping scenario, and object size (Fig. 1). Following conventional wisdom, we predicted that Ospreys photographed clutching fish were more likely to display a zygodactyl ($2 \times 2$) toe configuration. Furthermore, under the assumption that zygodactyly enhances grip force or the probability of prey contact (cited above), we anticipated that larger object (prey) sizes, (but not necessarily perching substrates), would also elicit a $2 \times 2$ toe configuration.

## MATERIALS AND METHODS

We searched the World-Wide Web (predominantly Google Images (English)) for photographs of Ospreys interacting with prey or various substrates, using the following search terms: "osprey," "*P. haliaetus*," and combinations of the previous two terms with "clutching," "grasping," "nest," "fish," and "photos." We then moved on to searching both personal and professional websites, and then videos (where we took screenshots of appropriate footage). Finally, we moved on to different languages of Google and repeated the above. Two observers independently scored each foot of each Osprey in every image for the characteristics described below and in Table 1. A third independent observer served as a "moderator," by compiling the scores of the other two observers and resolving any disagreements. The three observers rotated among tasks, such that each one served as a moderator for one component of the data set or another. We screened the data set for duplicated image file names to avoid scoring the same individual Osprey foot twice. We also ordered images by size and dimension to guard against the possibility that duplicated images were uploaded with different file names. Nevertheless, we cannot exclude the possibility that the same individual Ospreys might have appeared in more than one distinct image. The inadvertent inclusion of duplicated images and/or individual Osprey feet (i.e., pseudoreplication) would certainly inflate type I error rates. However, we have no reason to suspect any biases in the likelihood of duplicated images with respect to toe configuration and grasping condition, in which case the magnitudes (if not the significance levels) of the relationships among variables should remain relatively unaffected.

Each Osprey pictured in an image constituted a "subject," and each foot pictured was a replicate in the analyses. We used generalized estimating equations (a repeated-measures form of logistic regression; SPSS version 22; IBM, Armonk, NY, USA), with image identity included as a subject variable, and foot identity (left or right) included as a within-subjects variable, for which we specified an unstructured correlation matrix. We treated toe configuration as an ordinal (logistic) response variable ranging between 1 (= 3 × 1) and 3 (= 2 × 2), in which 2 (= 2.5 × 1.5) constituted an intermediate configuration analogous to *Bock & Miller's (1959)* "ectropodactyl" foot type (Figs. 1B, 1C and 1F). We performed two series of analyses: one overall test to examine the effects of relative "object size" (ordinal variable ranging 0 (no object) to 4 (extra-large); Table 1) and "grasping scenario" (0 = nothing in feet, P = perched on substrate, G = grasping an object), as well as their interaction. Although we were not specifically interested in the effects of foot identity (left or right), we performed an additional test including "foot identity" as a fixed effect to screen for any footedness biases (*Allen et al., 2018*). We then followed this analysis with a more refined test on data including only cases of contact between foot and object or substrate. For this test, we included an additional nested effect of "contact condition" (F = fish, O = other object, T = tree, S = other substrate; Table 1) within grasping scenario (P vs. G), to determine whether the general types of objects or substrates grasped have any further effects on toe configuration within each of the two main grasping scenarios. We also added an additional variable, "footing," indicating whether grasping was

**Table 1  List of variables included in the analyses, along with descriptions of each category.**

| Variable/categories | Description | Additional notes/justification |
|---|---|---|
| Toe configuration | | Treated as an ordinal logistic response variable |
| 1 (3 × 1) | Anisodactyl (digits II–IV directed cranially, digit I directed caudally) | |
| 2 (2.5 × 1.5) | Transitional; digit IV mid-way between digits III and I | |
| 3 (2 × 2) | Zygodactyl (digits II and III directed cranially, digits I and IV directed caudally) | |
| Grasping scenario | | To test how overall grasping behavior affects toe configuration |
| Free-footed (0) | Foot was empty; Osprey may have been landing, taking off, or diving | |
| Grasping object (G) | Object visibly clutched by foot; usually during mid-flight | |
| Perching (P) | Osprey was apparently motionless, with foot open against substrate | |
| Contact condition | | Effect nested within grasping scenario, to determine whether the type of object/structure contacted within each scenario (G or P, above) affected toe configuration |
| No contact (0) | Foot not in contact with anything | |
| Fish (F) | Foot enclosed a fish; usually upon leaving the water or in mid-flight or landing | |
| Other object (O) | Foot enclosed something other than a fish; usually nesting material, occasionally the talons of other Ospreys | |
| Tree (T) | Foot was enclosed around a tree branch while Osprey was perched | Trees were distinguished from other perching substrates to account for Ospreys' tendencies to wrap their toes around branches, as opposed to standing flat-footed |
| Other substrate (S) | Foot was in contact with perching substrates other than a tree branch; usually a post, rock, or ground | |
| Object size | | Assessed visually, relative to the extent to which toes encircled the object |
| 0 | No object in foot | |
| 1 | Small/very small: foot encircled between 67% and ≥100% of object "diameter" | By "diameter" we refer roughly to the cross-sectional dimension of the grasped object |
| 2 | Medium: foot encircled between 34–66% of object "diameter" | |
| 3 | Large: foot encircled 33% of object or less of object "diameter" | |
| 4 | Extra-large: foot did not really "wrap" around the object at all (e. g., ground, nest surface) | |

*(Continued)*

 

| Variable/categories | Description | Additional notes/justification |
| --- | --- | --- |
| Foot identity | Left or right foot scored | Included as a within-subjects variable to account for covariation in the responses between feet |
| Footing | Whether object was grasped with one (1) or both (2) feet | Included specifically to test whether single-foot grasps were more apt to exhibit zygodactyly, perhaps to enhance purchase on objects when unaided by the other foot |

**Note:**
Statistical analyses were designed in such a way as to model the probability of zygodactyly (dependent variable) with each condition.

performed with one or both feet. For both sets of analyses, we began with full models (main effects and interactions) and successively removed non-significant interactions (by order of decreasing $P$-value) to obtain the most parsimonious final models. Significance was based on the Type III sums of squares, and an $\alpha = 0.05$.

## RESULTS

The 1,184 images of Osprey grasping behavior that we scored (Data S1) fell into five main categories: (1) flying with fish, perching (2) with and (3) without fish, (4) nest-building, and (5) pre-contact with prey or substrate. Of these, obscured visibility of the feet and casewise deletions from one or more missing variables resulted in 1,123 Osprey images of $n = 1,882$ feet, both in contact with objects and not, entered into the analysis. Overall, there was no significant interaction between object size and grasping scenario on toe configuration (Type III Wald Chi-square ($\chi^2$) test of model effects = 4.34, d$f$ = 2, $P = 0.114$). The effect of grasping scenario remained significant ($\chi^2 = 198.61$, d$f$ = 1, $P < 0.0001$), and the effect object size remained non-significant ($\chi^2 = 0.457$, d$f$ = 3, $P = 0.928$), after removing the non-significant interaction term from the model. The parameter estimates ($B$; Table 2) revealed that the probability of zygodactyly significantly increased for the "object grasping" and "nothing in foot" scenarios, compared to the "perching" scenario (Fig. 2). In particular, the odds that an osprey exhibited a zygodactyl toe configuration during flight were 5.7 times greater when pictured grasping objects, and 2.6 times greater when grasping nothing, than whilst perched. There was no significant effect of foot identity (confirming of the lack of foot lateralization in Ospreys found by *Allen et al. (2018)*), nor any interaction with object size or grasping scenario, on toe configuration (Table S1).

When considering object-contact cases only ($n = 1,503$ feet from 995 images), all main effects and interactions were significant (Table 3). Both interaction effects involving footing and object size reflect variation in responses between contact conditions within each perching and grasping scenarios (Fig. 3). In the former case, the interaction was due primarily to an increase in the probability of zygodactyly from dual- to single-foot grasping for fish, relative to the "other substrate" reference contact condition of perching ($B = 0.882 \pm 0.378$, d$f$ = 1, $P = 0.019$, Exp ($B$) = 2.42 [1.15–5.07, 95% CI]). The latter

**Table 2  Parameter estimates and test statistics from a generalized estimating equation (GEE) model.**

| Parameter | | $B$ | Standard error | Hypothesis test | | | Odds ratio Exp ($B$) | 95% CI Exp ($B$) | |
|---|---|---|---|---|---|---|---|---|---|
| | | | | Type III wald $\chi^2$ | df | $P$ | | Lower | Upper |
| Threshold | Toecode = 1 | 0.172 | 0.2226 | 0.595 | 1 | 0.440 | 1.187 | 0.768 | 1.837 |
| | Toecode = 2 | 0.773 | 0.2236 | 11.942 | 1 | 0.001 | 2.166 | 1.397 | 3.357 |
| Graspscen = 0 | | 0.963 | 0.2517 | 14.629 | 1 | 0.0001 | 2.619 | 1.599 | 4.289 |
| Graspscen = G | | 1.739 | 0.3139 | 30.678 | 1 | <0.0001 | 5.690 | 3.075 | 10.527 |
| Objsize = 1 | | 0.308 | 0.2472 | 1.550 | 1 | 0.213 | 1.360 | 0.838 | 2.208 |
| Objsize = 2 | | 0.046 | 0.2533 | 0.033 | 1 | 0.856 | 1.047 | 0.637 | 1.720 |
| Objsize = 3 | | 0.116 | 0.2719 | 0.181 | 1 | 0.671 | 1.123 | 0.659 | 1.913 |
| Graspscen = G × objsize = 1 | | −0.415 | 0.3548 | 1.367 | 1 | 0.242 | 0.660 | 0.330 | 1.324 |
| Graspscen = G × objsize = 2 | | 0.078 | 0.3661 | 0.045 | 1 | 0.831 | 1.081 | 0.528 | 2.216 |
| (Scale) | | 1 | | | | | | | |

**Note:**
Toe configuration (toe code; 1 = 3 × 1, 2 = 2.5 × 1.5, 3 = 2 × 2) was modeled as a function of grasping scenario (graspscen; free-footed, grasping, perched), object size (objsize; no object (0)—extra-large (4)), and their interaction (graspscen × objsize), for the complete data set ($n$ = 1,882 feet (of 1,123 Osprey images)). Categorical levels omitted from the list of parameters in the table served as reference categories.

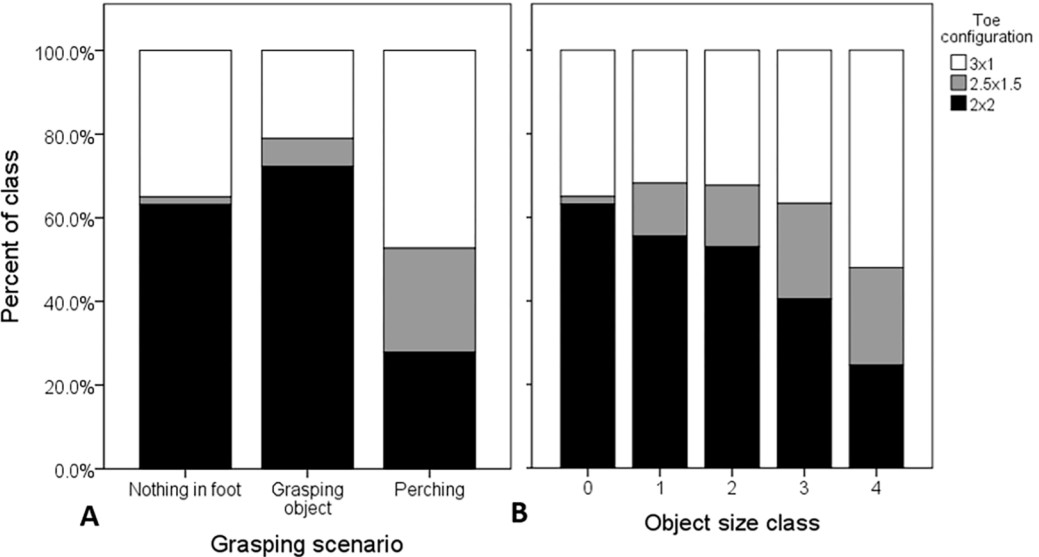

**Figure 2  Raw proportional distributions of toe configurations with respect to grasping scenario and object size, scored from 1,123 web images of Ospreys.** Toe configurations were classified as: 2 × 2 = zygodactyl, 3 × 1 = anisodactyl, and 2.5 × 1.5 = intermediate condition. The proportions of observations for each toe configuration across each grasping scenario (A), and relative object size class (B), were based on $n$ = 1,882 feet (left and right combined). When these variables were considered in the analysis simultaneously, the probability of zygodactyly (2 × 2) was significantly greater when Ospreys were photographed grasping objects, or nothing, than when perched, and there was no significant effect of object size.

interaction was due to two marginally non-significant ($P$ = 0.071–0.072) effects: a decrease in the probability of zygodactyly for small object sizes relative to large when grasping fish compared to the "other substrate" reference condition of perching, and an increase in the probability of zygodactyly for medium object sizes relative to large, when perched in

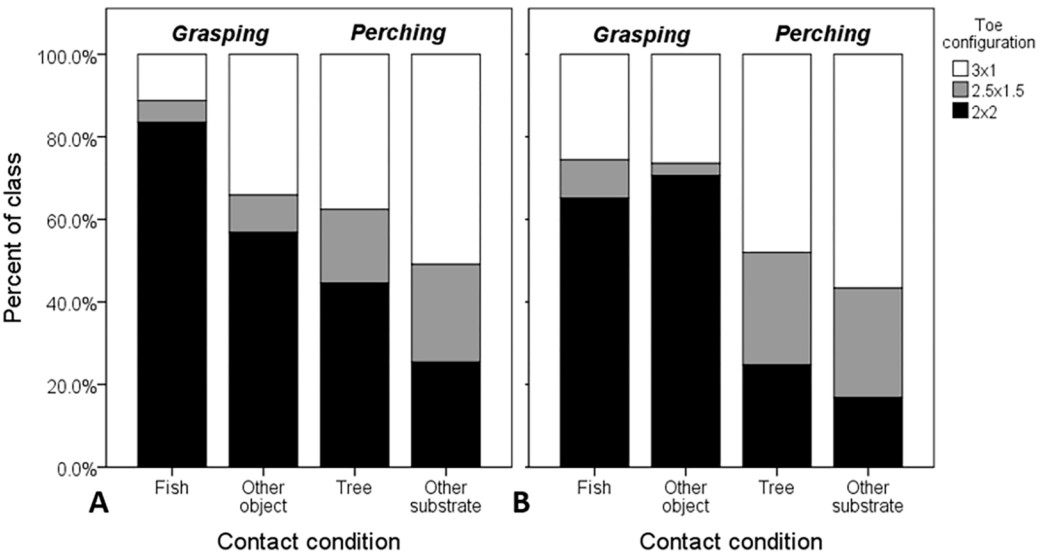

**Figure 3** **Raw proportional distributions of each toe configuration scored from 995 web images of Ospreys for single- and dual-foot grasps.** Toe configurations were classified as: $2 \times 2$ = zygodactyl, $3 \times 1$ = anisodactyl, and $2.5 \times 1.5$ = intermediate condition. Single-foot (A) and dual-foot (B) cross-tabulations with respect to grasping scenario and contact condition were based on $n = 1,503$ feet. When these variables were considered in the analysis simultaneously, the probability of zygodactyly ($2 \times 2$) was significantly greater, overall, when Ospreys were photographed grasping compared to perching, and specifically for single-foot grasps of fish compared to other objects, and trees compared to other substrates.

trees compared to "other substrates." However, because these parameters were not significant, we felt justified in excluding the object size × contact condition within grasping scenario interaction effect in subsequent analyses (below).

In the subsequent model, all effects remained significant, with the exception of object size (Table 3). Because the effect of contact condition within grasping scenario depended upon whether or not the grasp was single- or dual-footed, we generated new models for dual-footed ($n = 962$) and single-footed ($n = 541$) grasps, separately (Fig. 3). In both models the main effect of grasping scenario was significant (Table 3), such that the odds of zygodactyl grasps were 2.8 and 6.4 times greater during flying than perching (Table 4). Furthermore, there was a significant effect of contact condition within grasping scenario for single-footed grasps, but not for dual-footed grasps (Table 3). For the former, the probability of zygodactyly was significantly greater for the fish, compared to the "other object" contact condition, as well as for the tree, compared to the "other substrate" contact condition (Table 4).

## DISCUSSION

We analyzed grasping behavior of Ospreys from 1,184 web images and videos of Ospreys in various states of using their feet. Our results support predictions from casual observations, photographs, and anecdotal reports from the literature: that Ospreys tend to employ a zygodactylous foot configuration when grasping objects, and in particular when gripping fish. This suggests a functional association between predatory behavior and

**Table 3 Test of model effects from generalized estimating equation (GEE) models restricted to cases in which feet were observed contacting objects or substrates ($n$ = 1,503).**

| Source | Type III wald $\chi^2$ | d$f$ | $P$ |
|---|---|---|---|
| Graspscen | 23.68 | 1 | <0.0001 |
| Objsize | 8.33 | 3 | 0.040 |
| Footing | 5.20 | 1 | 0.023 |
| Graspcond (graspscen) | 18.68 | 2 | <0.0001 |
| Footing × graspcond (graspscen) | 18.58 | 3 | 0.0003 |
| Objsize × graspcond (graspscen) | 18.27 | 7 | 0.011 |
| Reduced model | | | |
| Graspscen | 98.86 | 1 | <0.0001 |
| Objsize | 0.464 | 3 | 0.927 |
| Footing | 5.25 | 1 | 0.022 |
| Graspcond (graspscen) | 15.29 | 2 | <0.0001 |
| Footing × graspcond (graspscen) | 16.38 | 3 | 0.001 |
| Footing = single-footed | | | |
| Graspscen | 27.95 | 1 | <0.0001 |
| Objsize | 0.339 | 3 | 0.952 |
| Graspcond (graspscen) | 18.30 | 2 | <0.0001 |
| Footing = dual-footed | | | |
| Graspscen | 86.66 | 1 | <0.0001 |
| Objsize | 0.791 | 3 | 0.852 |
| Graspcond (graspscen) | 1.92 | 2 | 0.383 |

**Note:**
Toe configuration (toe code; 1 = 3 × 1, 2 = 2.5 × 1.5, 3 = 2 × 2) was modeled as a function of grasping scenario (graspscen; free-footed, grasping, perched), contact condition (contcond; F, fish; O, other object; T, tree; S, other substrate) within grasping scenario, object size (objsize; small (1)—extra-large (4)), and footing (dual- or single-foot grasps). The reduced model shows results after excluding an interaction term with marginally non-significant parameter estimates; this model was further decomposed into separate models for each single ($n$ = 541) and dual ($n$ = 962) footing condition.

zygodactyly and has implications for the selective role of predatory performance in the evolution of zygodactyly more generally. Notably, the use of a zygodactylous configuration during single-foot grasps of fish (Fig. 3) strongly suggests that this toe configuration affords a performance advantage under the most challenging grasping conditions. Along these lines, however, it seems odd that object size was ostensibly unrelated to zygodactyly (Fig. 2), with a (non-significant) tendency for zygodactyl toe configurations to be pictured with smaller object sizes. On biomechanical grounds, very large and very small objects (relative to grasper size) pose greater challenges for grasping (*Seo & Armstrong, 2008*; *Irwin & Radwin, 2008*; *Fok & Chou, 2010*). Perhaps this is explained by the potential benefits of the multiarticular nature of their digital flexion mechanism (*Backus et al., 2015*), which might afford the ability to grasp a wide range of object sizes regardless of toe configuration (*Dollar & Howe, 2011*).

Embryological evidence supports developmental mechanisms as the primary drivers of toe configuration across taxa (*Botelho et al., 2014*; *Botelho, Smith-Paredes & Vargas, 2015*). Semi-zygodactyly appears in four avian clades: Ospreys, turacos, the common ancestor of owls and mousebirds (*Botelho, Smith-Paredes & Vargas, 2015*), and in ancestral

**Table 4 Parameter estimates and test statistics from generalized estimating equation (GEE) models for single-footed (*n* = 541) and dual-footed (*n* = 962) contact cases.**

| Parameter | | B | Standard error | Hypothesis test | | | Odds ratio Exp (B) | 95% CI Exp (B) | |
|---|---|---|---|---|---|---|---|---|---|
| | | | | Type III wald $\chi^2$ | df | P | | Lower | Upper |
| Single-footed grasps | | | | | | | | | |
| Threshold | Toecode = 1 | 0.093 | 0.3531 | 0.070 | 1 | 0.792 | 1.098 | 0.549 | 2.193 |
| | Toecode = 2 | 0.731 | 0.3549 | 4.245 | 1 | 0.039 | 2.077 | 1.036 | 4.165 |
| Graspscen = G | | 1.025 | 0.4664 | 4.826 | 1 | 0.028 | 2.786 | 1.117 | 6.950 |
| Objsize = 1 | | −0.064 | 0.4638 | 0.019 | 1 | 0.890 | 0.938 | 0.378 | 2.327 |
| Objsize = 2 | | −0.085 | 0.4461 | 0.036 | 1 | 0.850 | 0.919 | 0.383 | 2.203 |
| Objsize = 3 | | −0.193 | 0.4261 | 0.204 | 1 | 0.651 | 0.825 | 0.358 | 1.902 |
| Contcond = F (graspscen=G) | | 1.400 | 0.3793 | 13.619 | 1 | 0.0002 | 4.054 | 1.928 | 8.527 |
| Contcond = T (graspscen = P) | | 0.666 | 0.3270 | 4.145 | 1 | 0.042 | 1.946 | 1.025 | 3.694 |
| (Scale) | | 1 | | | | | | | |
| Dual-footed grasps | | | | | | | | | |
| Threshold | Toecode = 1 | 0.373 | 0.3168 | 1.389 | 1 | 0.239 | 1.453 | 0.781 | 2.703 |
| | Toecode = 2 | 1.208 | 0.3208 | 14.188 | 1 | 0.0002 | 3.347 | 1.785 | 6.277 |
| Graspscen = G | | 1.849 | 0.3149 | 34.456 | 1 | <0.0001 | 6.352 | 3.426 | 11.775 |
| Objsize = 1 | | 0.091 | 0.3589 | 0.064 | 1 | 0.800 | 1.095 | 0.542 | 2.213 |
| Objsize = 2 | | −0.031 | 0.3538 | 0.007 | 1 | 0.931 | 0.970 | 0.485 | 1.940 |
| Objsize = 3 | | 0.122 | 0.3735 | 0.107 | 1 | 0.744 | 1.130 | 0.543 | 2.349 |
| Contcond = F (graspscen = G) | | −0.142 | 0.2706 | 0.276 | 1 | 0.599 | 0.867 | 0.510 | 1.474 |
| Contcond = T (graspscen = P) | | 0.296 | 0.2257 | 1.722 | 1 | 0.189 | 1.345 | 0.864 | 2.093 |
| (Scale) | | 1 | | | | | | | |

Note:
Toe configuration (toe code; 1 = 3 × 1, 2 = 2.5 × 1.5, 3 = 2 × 2) was modeled as a function of grasping scenario (graspscen; free-footed, grasping, perched), contact condition (contcond; F, fish; O, other object; T, tree; S, other substrate) within grasping scenario, and object size (objsize; small (1)—extra-large (4)). Categorical levels omitted from the list of parameters in the table served as reference categories.

accipitrid kites (*Tsang, 2012*), most likely having arisen independently in some of these lineages (*Ksepka, Stidham & Williamson, 2017*). In most of these cases, semi-zygodactyly occurs in groups related to fully-zygodactylous clades, suggesting semi-zygodactyly as a potential intermediate, ancestral condition (*Botelho, Smith-Paredes & Vargas, 2015*; *Ksepka, Stidham & Williamson, 2017*). However, semi-zygodactylous Ospreys (Pandionidae) are nested well within the predominantly anisodactylous Accipitriformes (*Hackett et al., 2008*; *Yuri et al., 2013*; *Jarvis et al., 2014*; *Prum et al., 2015*), which, coupled with their extreme piscivorous specialization, suggests a possible adaptive role for semi-zygodactyly in this group. Furthermore, a recent analysis of the pedal flexibility of Australian raptors, including the Osprey, has indicated that diurnal raptors do indeed possess a wide range of angle divarication of digits (i.e., the degree to which toes are splayed out from one another) as a group (*Tsang & McDonald, 2018*). The Osprey exceeded the maximum digit angle divarication of digit IV (the digit that enables semi-zygodactyl grasping) of other anisodactylous raptors, achieving wider digit IV angle divarication results that overlapped with the digit IV angle divarications of the nocturnal owls. This degree of convergence between Ospreys and owls lends further support to

possible ecological, adaptive, origins of semi-zygodactyly, because Ospreys and owls feed mostly on prey that can be difficult to capture (e.g., Ospreys plunge-diving for slippery fish and owls nocturnally hunting fast-moving small mammals). Both groups rely mostly on stealth to snatch their prey unawares from under the cover of water or darkness, respectively. However, both groups also possess morphological and behavioral modifications presumed to meet the potential added challenges of their prey. For example, both groups lack a stretch of skin between digits III and IV typical of other raptors, which facilitates a wider lateral movement of digit IV. Furthermore, previous researchers have cited the relatively strong gripping forces of owls compared to diurnal raptors, as a mechanism for overcoming the relatively greater difficulties they experience hunting nocturnally (*Marti, 1974*; *Ward, Weigl & Conroy, 2002*; *Einoder & Richardson, 2007*).

However, there are other taxa that are semi-zygodactylous (e.g., mousebirds and turacos), zygodactylous (e.g., parrots, woodpeckers, and roadrunners), and heterodactylous (trogons) that do not capture prey with their feet as do Ospreys and owls, and there are several other anisodactylous species that do (e.g., falconiforms and most accipitriforms) (*Raikow, 1985*). Although functional, adaptive arguments have been made for some of these (e.g., for climbing or manipulating food with the toes; *Bock & Miller, 1959*; *Berman & Raikow, 1982*), definitive conclusions await more comprehensive, phylogenetically-informed analysis. Nevertheless, developmental mechanisms and ecological factors are not mutually exclusive, and it is likely that different combinations of these factors influence the evolution of semi-zygodactyly across taxa.

The ability to transition between toe configurations is a feat of which very few species are capable. We present quantitative data linking prey capture behavior with zygodactyly in Ospreys. Nevertheless, the extent to which semi- or full-zygodactyly provides a distinct performance advantage for grasping has yet to be explicitly tested. *Bartosik (2009)* suggested that the flexibility afforded by the ability to reverse the outer toe helps Ospreys optimize their grasps on the lateral sides of fish so as to avoid contact with their sharp dorsal spines. Thus, further work is required, supported by consistent field observations of reliably located individuals at close range, to facilitate further study of this unique behavior. Citizen science potentially has much to offer in this regard, via nest cams or automated cameras positioned near prime foraging grounds (*Bierregaard, Poole & Washburn, 2014*). Another important avenue of inquiry currently underway is to uncover precisely how rotation of the outer toe is controlled; that is, whether it is driven entirely by the action of M. abductor digit IV, or through joint coupling mechanisms facilitated by the morphology of the tarsometatarso-phalangeal joint and other multiarticular digital tendons.

## CONCLUSIONS

From our analysis of web images, we found that semi-zygodactylous Ospreys are pictured using three predominant toe configurations: anisodactylous, zygodactylous, and an intermediate condition we labeled "2.5 × 1.5". Our generalized estimating equation models confirmed the oft-cited association between zygodactyly and grasping behavior in general; the odds that an osprey exhibited zygodactyly while pictured grasping objects in

flight were 5.7 times greater than whilst perched. Contrary to our expectations, zygodactyly was unrelated to object size, but the odds of observing zygodactyly in single-foot grasps were 4.1 times greater with fish compared to other objects. This suggests a functional association between predatory behavior and zygodactyly, and ultimately has implications for the selective role of predatory performance in the evolution of zygodactyly.

## ACKNOWLEDGEMENTS

We kindly thank Hans Kassier, Chad Small, and Jon Simmonds for their hard work and countless hours accessing and scoring digital web images of Ospreys. We are very grateful to Julieta Carril, Richard O. Bierregaard, and another anonymous reviewer, whose constructive comments greatly improved the final version of this paper.

### Funding
The authors received no funding for this work.

### Competing Interests
The authors declare that they have no competing interests.

### Author Contributions
- Diego Sustaita conceived and designed the experiments, performed the experiments, analyzed the data, prepared figures and/or tables, authored or reviewed drafts of the paper, approved the final draft.
- Yuri Gloumakov performed the experiments, analyzed the data, prepared figures and/or tables, authored or reviewed drafts of the paper, approved the final draft.
- Leah R. Tsang prepared figures and/or tables, authored or reviewed drafts of the paper, approved the final draft.
- Aaron M. Dollar conceived and designed the experiments, performed the experiments, contributed reagents/materials/analysis tools, authored or reviewed drafts of the paper, approved the final draft.

### Data Availability
The data used for the analysis, tables, and figures are available as Supplemental Data S1.

### Supplemental Information
Supplemental information for this article can be found online at http://dx.doi.org/10.7717/peerj.6243#supplemental-information.

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
