# Peer review of "Behavioral correlates of semi-zygodactyly in Ospreys (Pandion haliaetus) based on analysis of internet images"

_PeerJ, doi:10.7717/peerj.6243_

## Round 0.1 · original submission · Minor Revisions

Hi Diego
sorry for the delay in make my decision. I have now three reviews and all of them suggest a number of points that could improve the good work done so far. Although the paper requires minor changes for it to be accepted, please take all suggestions in full consideration.

·

Basic reporting

I'm not a native English speaker, but the manuscript seems to be written in clear, unambiguous and technically correct English.
The literature is well referenced, exhaustive and appropriate.
The structure of the article fits to the PeerJ journal standards.
Figures are relevant and of high quality (some minor points regarding Figures 1 and 3 are detailed in the pdf file).

Experimental design

The article suits with the aims and scope of the PeerJ journal.
The research question is well defined and meaningful, since the article seeks to rigorously test the association between facultative zygodactyly of the Osprey with capture of the prey, a statement that so far was based on anecdotal observations.
Methods are described with sufficient detail.

Validity of the findings

The results are well presented and conclusions are well stated.
These are interesting findings and deserve to be published.

Additional comments

This is a novel and interesting article where the authors, based on internet images, cross-tabulated the three possible toe configurations for the Osprey (i.e. anisodactylous, zygodactylous, and the intermediate condition) with different grasping scenarios, contact conditions, object sizes and grasping behaviours. Their results confirms an association between zygodactyly and grasping behaviour in general for the Osprey, with greater odds of exhibiting zygodactyly while grasping objects in flight than whilst perched and in single-foot grasps with fish compared to other objects, suggesting a functional association between predatory behaviour and zygodactyly. Minor specific suggestions are noted in the manuscript pdf.

·

Basic reporting

The paper meets all these basic requirements. All papers in the literature cited are in the text. I did not do the reverse check to confirm that all citations in the text of the paper are in the LitCit.

Experimental design

This is an interesting topic and an admirable use of the amazing resources available in the form of thousands and thousands of digital images available online.

I find the methods for the most part strong and well thought out. One question that is not addressed, and I don’t really think it’s functionally important, is whether Ospreys are “right-“ or “left-footed” when they’re carrying prey. It is well known that Ospreys will maneuver fish once they’re airborne with their prey so that the fish’s body is aligned with the Osprey’s body to turn the fish’s hydrodynamic design into an aerodynamic advantage as they carry the prey away from the capture location. I get asked this all the time. My guess is that in most cases it is just a function of whether the fish was going from right to left when the Osprey caught it. Whichever foot is closest to the head will be the one that the Osprey puts in front. Just an observation for future analysis perhaps.

One bit of data that is certainly available in some images is what was the toe arrangement before the Osprey hit the water? Kris Rowe has amazing images of Ospreys about to strike the water and they all seem to have the toes fully splayed in the zygodactylous arrangement. Are there enough images like these to offer any insight here? The authors must have included many of Rowe's photos from the Connecticut River Ospreys.

Validity of the findings

I will have to defer to other reviewers on the statistical analyses. I’m not familiar with the procedures that the authors used, but they seem relatively straightforward, and the results make sense, so I will move to the discussion and general comments.

Lines 59-62. I don’t follow the logic here. Sure, we see Ospreys using a zygodactyl toe arrangement doing non-predatory behavior, but wouldn’t that just be expected given that the outer toe can go either way? Why does seeing an Osprey perched zygodactyly make arguments about the advantages of zygodactyly in prey capture speculative? Logic dictates that the selective forces on a raptor’s foot will be related to prey capture. I demonstrated this in my Ph.D. thesis—foot structure (specifically toe length) across the Falconiformes and Accipitriformes is tightly linked to the type of prey commonly taken by different species.

Lines 199-200. The authors state that Semi-zygodactyly has “apparently evolved only three times (Ospreys, turacos, and the common ancestor of owls and mousebirds),…” but on line 51 they state that Black-shouldered Kites are also facultatively zygodactylous. If the Kites have the trait, then the statement on lines 199-200 cannot be true.

Lines 211-214. I disagree with the statement about the similarity of owls and Ospreys relating to their feeding on prey that is difficult to capture driving some sort of convergent evolution. No doubt about Ospreys needing every advantage they can get holding on to slippery fish, but the same is certainly not relevant to owls. Most owls that are hunting at night are not capturing fast moving prey. Most are catching prey that are almost certainly unaware of what is about to happen to them. I would argue that diurnal raptors are much more likely to encounter fast moving prey than are most owls. The authors need to make a much stronger case for this point.

Lines 226-229. The authors question how the rotation of the outer toe is accomplished, suggesting an alternative that the arrangement is “passively enabled by contact.” It took me a while to see the point here, I think because of the use of “enabled.” An Osprey about to grab a fish has set the toes in whatever arrangement it chooses before the moment of impact. I can attest to the same situation with Barred Owls. In my studies of this species, I have been hit by owls and can see, looking at the punctures in my arms, that they were in fully zygodactyl foot arrangement when they sunk their talons into me. As evidenced by the photos I’ve seen and mentioned above, Ospreys have their talons fully splayed at the moment of moment of impact. I can see how the movement of a fish just before impact could RESULT IN (rather than "enable") an alternate toe position, or perhaps more likely, this could happen as an Osprey, after leaving the water, regrasps its prey to manipulate it into the head-first orientation. This possibility should be discussed.

It would be interesting to work with a dead Osprey and manipulate the legs in the full “jack-knife” position that they assume before hitting the water to see if the toes splay out automatically, the way toes automatically clench around a perch when the taro-metatarsal joint is flexed.

In Table 1, right column in Grasping Effects, “To test how overall grasping behavior effects toe configuration,” change “effects” to “affects.”

Additional comments

I've gone through a Word version of the file and made some comments via TrackChanges. Most are minor points of grammar or style. Most of my more significant comments are above. I'm going to insert a couple of pictures of Ospreys immediately pre-water contact to illustrate the point about toe arrangement being determined will before contact with prey.

Reviewer 3 ·

Basic reporting

No comments

Experimental design

No comments

Validity of the findings

no comments

Additional comments

The authors recollected photos of osprey from the internet and scored the position of digit four in relation to the observed behavior. The conclusion is not surprising, but this work quantifies, for the first time, something others have notice before, namely that ospreys fish with zygodactyl feet, but perch in anisodactyl position.
The manuscript is well written, the introduction is especially clear.
The results section, on the other hand, is hard to read, and it would be improved if the authors could soften it reducing the amount of raw data in the text.
The methodology is unusual, but very welcome, and it is actually what I like most in the paper. There is a huge amount of data on natural history to be explored by scientists in the web. They have shown an example of how to do it.
The discussion points to adaptative implications for the origin of semi- and full zygodactyly. There are many counter examples that could be discussed to make the discussion more balanced. Semi-zygodactyl Turacos are not carnivorous, and some of the best fishing eagle are anisodactyl. The are some important new papers on zygodactyl and semi-zygodactyl fossil birds that could be included in the discussion (see works from Daniel Ksepka, and Gerald Mayr).

Minor comments:

1 "Despite the common knowledge of Osprey semi-zygodactyly" This could be supported by some citations, it would be good to see what previous authors have said.
2 The sentence starting in lines 64-68 could be split in two for the sake of clarity.
3 194 “nested well within the predominantly anisodactylous Accipitriformes”. Actually Ospreys are the sister groups of all other Accipitriformes.

---

## Round 0.2 · accepted · Accept

Thank you for carefully addressing all the reviewer comments in your revised version. I think that your manuscript is ready to publish.

#